# Hierarchical Structure of Protein Sequence

**DOI:** 10.3390/ijms22158339

**Published:** 2021-08-03

**Authors:** Alexei N. Nekrasov, Yuri P. Kozmin, Sergey V. Kozyrev, Rustam H. Ziganshin, Alexandre G. de Brevern, Anastasia A. Anashkina

**Affiliations:** 1Shemyakin-Ovchinnikov Institute of Bioorganic Chemistry, The Russian Academy of Sciences, Miklukho-Maklaya St. 16/10, 117997 Moscow, Russia; alexei_nekrasov@mail.ru (A.N.N.); zibotic@mail.ru (Y.P.K.); ziganshin@mail.ru (R.H.Z.); 2Steklov Mathematical Institute and of Russian Academy of Sciences, 8 Gubkina St., 119991 Moscow, Russia; kozyrev@mi.ras.ru; 3INSERM UMR S-1134, DSIMB, Univ. Paris, INTS, Lab. of Excellence GR-Ex 6, rue Alexandre Cabanel, CEDEX 15, 75739 Paris, France; alexandre.debrevern@univ-paris-diderot.fr; 4Engelhardt Institute of Molecular Biology, Russian Academy of Sciences, Vavilov St. 32, 119991 Moscow, Russia

**Keywords:** protein structure, hierarchy, protein sequence, ANIS method, super secondary structure

## Abstract

Most non-communicable diseases are associated with dysfunction of proteins or protein complexes. The relationship between sequence and structure has been analyzed for a long time, and the analysis of the sequences organization in domains and motifs remains an actual research area. Here, we propose a mathematical method for revealing the hierarchical organization of protein sequences. The method is based on the pentapeptide as a unit of protein sequences. Employing the frequency of occurrence of pentapeptides in sequences of natural proteins and a special mathematical approach, this method revealed a hierarchical structure in the protein sequence. The method was applied to 24,647 non-homologous protein sequences with sizes ranging from 50 to 400 residues from the NRDB90 database. Statistical analysis of the branching points of the graphs revealed 11 characteristic values of y (the width of the inscribed function), showing the relationship of these multiple fragments of the sequences. Several examples illustrate how fragments of the protein spatial structure correspond to the elements of the hierarchical structure of the protein sequence. This methodology provides a promising basis for a mathematically-based classification of the elements of the spatial organization of proteins. Elements of the hierarchical structure of different levels of the hierarchy can be used to solve biotechnological and medical problems.

## 1. Introduction

The multiplicity of functional characteristics of proteins is associated with a wide variety of their spatial structures. This diversity is ensured by the different arrangement of amino acid residues in their sequences. Hierarchical organization in the spatial structure of proteins was revealed by using various approaches such as, for example, calculating the local packing density of atoms in the structure [1,2,3,4], interaction energy inside a protein globule [5], Fuzzy Oil Drop sites [6] or hydrophobic folding nuclei [7]. Such hierarchical elements of proteins were considered as elements of protein folding at different stages [8,9,10,11]. However, in contrast to the structural elements obtained in the analysis of spatial structures, protein sequences have not previously been found to have a hierarchical structure. 

In this paper, we considered proteins as a system of hierarchically organized elements. In the spatial structure, elements at the top level of the hierarchy are structural domains [1,9]. Structural domains have almost all the features of natural polypeptide chains and, first of all, the ability to self-assemble. Obviously, these features should be visible in the protein sequences. Early studies of protein sequences showed that the complexity of the sequences of natural proteins differs from random polypeptides of the same amino acid composition by approximately 1% [12]. This leads to the representation that the protein sequences are “slightly edited random sequences” [13,14]. Obviously, such concepts do not correspond to the observed structural and functional properties of proteins as seen with the large number of approaches to predicting coding zones in genomes by Hidden Markov Model approaches [15]. To resolve this contradiction, we proposed a new paradigm [16,17,18]. We have shown [16] that a low level of Shannon informational entropy [19] is observed in a range of two to eight amino acid-fragment; the lowest level is observed for pentapeptides. Taking a fragment of five amino acid residues as a unit of protein sequence, we proposed the ANIS (ANalysis of Informational Structure) method for identifying hierarchically organized structures in protein sequences [17,18]. This method identifies tree-like hierarchical structures (graphs) in a protein sequence. Several examples have shown that free-standing graphs correspond to structural domains [20]. The ANIS method was confirmed in a number of experimental researches [20,21,22]. Subsequently, we used the revealed hierarchical elements for protein design [23,24,25,26] and for the study of the mechanisms of protein function [20,21,22]. It was experimentally shown [22,23,25,26] that the removal of sequence fragments corresponding to free-standing graphs from the native protein sequence leads to minimal folding distortion of the recombinant protein. In [20,22], using the analysis of hierarchical elements in the protein structure, mechanical models of functioning of protein molecular machines were proposed. 

An approach in which a protein sequence is considered as a system of blocks of five amino acid residues was used in the literature to create a protein structural alphabet [27], to study topologically stable elements of protein spatial structure of the lowest level [28], and to describe the folding of protein molecules [29,30]. In 2020, Kaushik and Zhang proposed a method for distinguishing between natural and random protein sequences, which takes into account the occurrence of amino acids and tripeptides [31]. As we have shown earlier [16], low level of informational entropy is observed in a range of two to eight amino acid-fragment; the lowest level is observed for pentapeptides. We assume that the use of pentapeptides instead of tripeptides should improve the quality of natural/random sequence recognition. 

Application of the ANIS method to a large number of natural protein sequences has led us to observe that there are characteristic sizes of sequence fragments in which tree-like graphs are divided into smaller hierarchical elements. The size of the fragments is smaller than structural domains, but larger than the structural alphabet proposed by de Brevern [27]. This work is devoted to the analysis of sizes of the revealed hierarchically organized elements of the protein sequence. 

## 2. Results

### 2.1. Correllation between ELements of Informational Structure (ELIS) of Different Hierarchical Levels

The correlation matrix $\left(r_{yy{}^{\prime }}\right)$ contains the data on correlations between ELIS (hierarchical ELements of Informational Structure) of different values of y and $y^{\prime }$ in protein hierarchical structure (Equation (9)). Correlation matrix $\left(r_{yy{}^{\prime }}\right)$ was built for 24,647 protein sequences with sizes in between 50 to 400 of amino acid residues from the NRDB90 database [32]. 

Figure 1A–I shows a graphical representation of the matrix elements $r_{yy{}^{\prime }}$ calculated by Equation (9). This matrix reflects agreements of branching points for different y values. Values of matrix elements of $\left(r_{yy{}^{\prime }}\right)$ lie in the interval from 0 to 1. Matrix elements equal to 1 are shown in black and located only on the diagonal ($y=y{}^{\prime })$. Other matrix elements are displayed in the figure in gray, if they exceed the specified threshold value. The threshold value is shown above the corresponding figure.

If branching points in hierarchical structures are situated at close y values, then matrix $\left(r_{yy{}^{\prime }}\right)$ will contain nonzero elements around the diagonal (see Figure 1). These non-zero elements (the red squares in the figure) form continuous square areas (CSA). 

Let us look at how the gray elements change with decreasing of threshold value. If there is a region of neighbor gray cells, then a CSA is formed in the diagonal region. It is highlighted in red. Let us reduce the threshold value. The number of gray cells will increase and they will form CSA in different places and of a greater size. At 0.01 of threshold value (Figure 1H), comes a moment that new gray cells do not appear, new CSA in the diagonal area are not formed. Thus, the structure of CSAs in the diagonal region became stable.

Table 1 shows the sizes and positions of the stable CSA. For the set of samples of non-homologous protein sequences analyzed in this work, these near-diagonal CSA correspond to the levels of hierarchical organization in the protein sequences, and, therefore, in the spatial structure of proteins. We assumed that there is a relationship between the levels of organization in the protein sequence and its spatial structure.

### 2.2. Hierarchy in the Spatial Structure of Proteins

The example below illustrates the relationship between hierarchical elements of the sequence and the corresponding fragments of the spatial structure. Figure 2A shows the hierarchical structure of the protein sequence of the photosynthetic reaction center from Rhodopseudomonas viridis (PDB id 1PRC, chain C) obtained using the ANIS method.

The architecture of N- and C-terminal ELISes can be considered qualitatively if they are approximated beyond the boundaries of the triangular domain of the function $H_{I}(x,y)$ definition. Continue the branches in the same direction when they reach the border of the triangular area.It can be seen that the ELIS located at the N- and C-ends of the sequence do not tend to merge with the ELIS located in the center of the sequence. Therefore, we can assume that the elements of the spatial structure at the N- and C- ends of the sequence are weakly correlated with the ELIS in the central part of the protein.

In the central region of the sequence, between positions 90 and 248, two independent ELIS are defined (green and red) (Figure 2A). The boundary between them is the residue 121. It should be noted that ELIS, formed between 121 and 248 residues (red color), has a very complex structure; the branches forming it merge at y values of more than 100. Figure 2A shows the spatial structural elements of the protein corresponding to ELIS also. First ELIS (green color) corresponds to α-helix with loop-shaped structures at its ends. Second ELIS, highlighted in red, has complex spatial structure. 

Next, consider spatial structure that corresponds to ELIS 121–248 (Figure 2B). The first element 121–135 is a small loop-like structure that terminates the α-helix and provides turn of the polypeptide chain (marked in red in Figure 2B). The second element 135–162 forms the β-hairpin and N- and C-turns of adjacent polypeptide chain (marked green in the Figure 2B). The third element 162–192, marked blue, is a short α-helix with N- and C- turns of adjacent polypeptide chain. The last ELIS 192–248 is highlighted in violet (Figure 2B) and is the longest at this level of the hierarchy. Its spatial structure is quite complex and includes an α-helix and an irregular fragment of the polypeptide chain. 

The spatial structure of 192–248 ELIS is quite complex (Figure 2C and Figure 3). It consists of 6 lower-level ELIS, which are: red ELIS (192–200)—a fragment of the polypeptide chain terminates the α-helix and implements the turn of the polypeptide chain; orange ELIS (200–206)—a fragment of the polypeptide chain located in an extended conformation; cyan ELIS (206–218) is a fragment of a polypeptide chain that forms a Π-shaped loop with a helix turn located in the C-terminal part of the fragment; green ELIS (218–227)—a fragment of the polypeptide chain that forms the initiator of the α-helix; blue ELIS (227–234) is a fragment of the polypeptide chain that supports the α-helix conformation; violet ELIS (234–248) is a fragment of a polypeptide chain that sequentially forms the termination of a α-helix, reverse rotation, and initiation of the next α-helix.

## 3. Discussion

The relationship between the amino acid sequence of a protein and its spatial structure seems to us intuitively quite obvious. We know that the amino acid sequences of proteins are encoded by the corresponding genes, and evolutionarily related organisms have similar sequences of genes and proteins that perform the same functions. This means that such protein sequences fold into spatially similar structures, especially in the region of active sites. Moreover, quite a large number of amino acid substitutions are often required to change the topology of the protein fold. There are examples where the topology of the fold is preserved when the sequence similarity is less than 30%. However, there are other examples when a single substitution leads to a change in protein folding and causes severe pathology.

Let us take a look at the process of protein synthesis on the ribosome. Protein folding essentially begins during synthesis, when the protein is only partially synthesized. Folding depends on local interactions of side chains of closely spaced amino acid residues. Accordingly, for correct folding, local interactions between certain amino acid residues should occur at the right stage, which leads to the formation of the correct pre-folding conformation, i.e., some positions in the protein sequence must be correlated. We have shown [16] that such a cross-correlation of positions in the amino acid sequence actually exists. If we consider the merging points of the branches of hierarchical trees in a large set of sequences of natural proteins (Figure 1), we can see that there are characteristic values of the width of the inscribed function *y*, at which branching occurs (Table 1). These characteristic values of the width of the inscribed function *y* correspond to the values of the sizes of protein fragments. Each of these characteristic values requires additional careful consideration and analysis.

Analyzing hierarchical structures of proteins, we can summarize that ELIS of the upper levels of the hierarchy usually correspond to structural domains [20]. At the lower levels of the ELIS hierarchy, elements of the super-secondary structure (β-hairpins, α-hairpins), elements of the secondary structure (α-helixes, β-strands, or extended structures), as well as spatial elements that provide a transition from one known element of the spatial organization to another are found. Structures revealed at the lowest level using the ANIS method are the result of the molecular evolution of polypeptide chains, since the elements of the hierarchy are formed by more common pentapeptides [33,34] (see Fragments and hierarchical structure of a protein, Equation (1)). 

Earlier [28], we showed that there are stable pentapeptides among elements of the lowest level of the hierarchy. Conformationally stable pentapeptides have been classified into classes partially corresponding to elements like α-helices and β-sheets. In addition to the classical elements of the secondary structure, pentapeptide structures with the fixed topological direction were found that ensure the transition from one element of the secondary structure to another.

This means that not only the regular structural elements themselves in the spatial structures of proteins are important, but also the direction of further course of the polypeptide chain. Undoubtedly, this plays a very important role in the folding of the polypeptide chain. Note that the proposed method for identifying structural elements allows us to reveal new classes of structural elements. Some of the elements detected by ANIS method do not correspond to usual structure classification elements such as α-helices and β-sheets (see description for Figure 2C and Figure 3 in the text). A distinctive feature of the ANIS method is the recognition of irregular, “non-classic” structural elements of the polypeptide chain. 

## 4. Materials and Methods

### 4.1. Fragments and Hierarchical Structure of a Protein

As mentioned earlier, the greatest self-consistency is observed within blocks of five amino acid residues [16]. Let us consider fragments of length five in proteins. We consider all possible overlapping fragments, i.e., neighboring fragments overlap with four residues. To any fragment A of length five from the protein sequence, we put in correspondence the frequency—the number φ(A) of occurrence of the fragment in the database of non-homological protein sequences. We will consider the total frequency: (1)$$\Phi (A)=\sum_{J:d(A,J)\leq \delta }\varphi (J)$$
over sequences J of length five with Hamming distance from A not larger than δ=1. In our papers [17,18], Hamming distance δ equal to one was also used, i.e., we summarized over fragments which differ from the initial fragment at no more than one amino acid residue. 

We will put in correspondence to a protein sequence (as a sequence $I=i_{1}\ldots i_{N}$ of amino acid residues) a sequence of fragments $I_{j}$ (pentapeptides) enumerated by central residues in fragments, i.e., j=3,…,N−2. Let us put in correspondence to a j-th pentapeptide in the protein I a value given by (1). Here, we sum over pentapeptides which differ from $I_{j}$ in no more than one residue (Hamming distance δ not larger than one) and frequencies are taken from the database NRDB90. Let us introduce for a protein with sequence I the function $f_{I}(j)=\Phi (I_{j})$ of total frequency of pentapeptides. 

Let us consider the Gaussian function on real axis (as in [17,18]): (2)$$g(x,y)=\frac{1}{y\sqrt{2\pi }}e^{-\frac{x^{2}}{2y^{2}}}$$
and let us introduce the smoothing distribution of total frequency of pentapeptides in a protein *I* as a convolution of the Gaussian function and the total frequency of pentapeptides in a protein: (3)$$F_{I}(x)=\sum_{j=3}^{N-2}f_{I}(j)g(x-j,y)$$

Then, let us fit Gaussian functions in the graph of the function $F_{I}(x)$, i.e., we will obtain the function: (4)$$H_{I}^{}(x,y)=\max h:\min z:[F_{I}(z)-he^{-\frac{{\left(z-x\right)}^{2}}{2y^{2}}}]\geq 0$$
which measures the height of Gaussian function with center in x and width 2y which can be fit in the graph of function $F_{I}(x)$. 

Function $H_{I}^{}(x,y)$ (4) defined in isosceles triangle in the coordinate plane (x,y) with the base x∈[1,N] at the abscissa axis and height y∈[1,N/2]. An example of the function $H_{I}(x,y)$ is shown at the Figure 4A. Local maxima of the function $H_{I}(x,y)$ (with respect to the abscissa when the ordinate is fixed) constitute a tree-like graph (as shown in Figure 4B). It is natural to put in correspondence branches of this graph to elements of hierarchical structure of protein sequence (namely ELIS, hierarchical ELements of Informational Structure [18], see Figure 4). 

Each hierarchical element is characterized by position in the protein sequence, number of branches contained in the element and the rank in the hierarchy. Let us consider one of hierarchical elements of the graph of $H_{I}(x,y)$ with branching at the point $\left(x_{0}.y_{0}\right)$. At this point, the graph branches into several elements with lower levels of the hierarchy (Figure 4B). 

Let us note that hierarchical structure of protein sequences can be very diverse. Figure 5 shows hierarchical structures for several protein sequences from UniProt database. 

### 4.2. Hierarchy in Structure of Protein Sequences

We selected non-redundant set of protein sequences from 50 to 400 amino acid residues from the NRDB90 database [32]. As a result of this selection, we obtained a set of 24,647 protein sequences. As it was shown in [16], a set of several thousand protein sequences is sufficient to reveal the regularities common to all proteins. The protein dataset must satisfy only two criteria: be large enough and not contain homologous sequences. 

For each protein from this set let us compute function $H_{I}(x,y)$ given by Equation (4). Lengths N of protein sequences vary from 50 to 400. 

For a protein I and fixed half-width y let us consider function $H_{I}^{norm}(x,y)=\frac{H_{I}(x,y)}{\int H_{I}(x,y)dx}$ as a probability distribution, i.e., let us normalize function $H_{I}(x,y)$ to satisfy
(5)$$\int H_{I}^{norm}(x,y)dx=1$$

Shannon entropy for this probability distribution has the form
(6)$$S_{I}(y)=-\int H_{I}^{norm}(x,y)\log H_{I}^{norm}(x,y)dx$$

For a partition of segment [0,1] in n equal segments (with length 1n), entropy of partition will be equal: (7)$$-\sum_{i=1}^{n}\frac{1}{n}\log\frac{1}{n}=-\log\frac{1}{n}=\log n$$

In this case, entropy depends only on number of fragments *n*. 

Let us consider the following difference between real entropy of protein sequence (6) and model entropy of distribution (7)
(8)$$\tilde{S_{I}}(y)=S_{I}(y)-\log n$$

This function has leap in points of branching of the tree-like graph where ELIS of smaller rank emerges. Now, let us calculate the derivative $\overset{}{\tilde{S{{}^{\prime }}_{I}}(y)}$ of the function $\overset{}{\tilde{S_{I}}(y)}$ with respect to y. Local maxima of this function will correspond to branching points of the graph. Figure 6B shows an example of hierarchical structure of a protein sequence and the derivative (Figure 6A). 

Next, let us consider for each fixed y=1,…,N a vector $V_{y}$ which contains derivatives $\overset{}{\tilde{S{{}^{\prime }}_{I}}(y)}$. This vector consist of I elements $\overset{}{\tilde{S{{}^{\prime }}_{I}}(y)}$ from each protein sequence of 24,647 dataset. Let us consider the correlation of these vectors for different y values: (9)$$r_{yy{}^{\prime }}=\frac{\langle V_{y},V_{y{}^{\prime }}\rangle }{\sqrt{\langle V_{y},V_{y{}^{\prime }}\rangle \langle V_{y{}^{\prime }},V_{y{}^{\prime }}\rangle }},\;\langle A,B\rangle =\sum_{I}A_{I}B_{I}$$

## 5. Conclusions

In this paper, a rigorous description of the method for studying of information in protein sequences is given—the method for analyzing the hierarchical information structure of protein sequences (ANIS method). We applied the ANIS method to 24,647 non-homologous protein sequences with sizes from 50 to 400 residues from the NRDB90 database [32] and identified elements that form a hierarchical structure in the protein sequence. We have shown that there is a correlation between the identified elements of different sizes. In this work, we revealed 11 characteristic values of smoothing function width value *y* for different CSA. In some way, the characteristic CSA values correspond to levels of structural organization in protein sequences. In addition, we have shown the relationship between the identified elements of the sequence and the elements of the spatial structure of proteins. 

The tremendous success of AlfaFold in predicting the structure of proteins from their amino acid sequence did not bring an understanding of the folding process. A deep neural network predicting the pattern of contacts between amino acids does not explain the principles of such prediction. Therefore, AlfaFold cannot be used to design a polypeptide chain with a desired topology. The approach that we propose in this article allows us to locate sites in the sequence responsible for the formation of the correct pre-folding conformation. We have previously assumed that the elements at the lowest level in the hierarchy have a number of topological constraints [28]. Thus, elements at the lowest level in the hierarchy can play a role of folding nuclei or make pre-folding conformation. Some amino acid substitutions can be similar for folding, but the introduction of inadequate substitutions can lead to the loss of the local topology of the polypeptide chain and disruption of folding in general. The method proposed here makes it possible to predict and assess the significance of the replacement.

Structural domains are stable structural elements and may independently form the 3D structure. Correspondence between the highest rank ELIS to structural domains may point to the important role of high rank ELIS in the formation of the spatial structure of proteins. We assume that the elements of 3D structure corresponding to the elements of information structure of other ranks are also stable elements of protein 3D structure. This method is promising for use in the design of recombinant proteins, since the removal or addition of individual branches of the hierarchical tree should not affect the folding of the protein as a whole. Previously, this method was already used for the design of recombinant proteins and solving biotechnological problems [21,22,23,24,25,26]. 

Our method also provides opportunities for studying the evolution of proteins. In this article, the value of δ in Formula (1) is in Section 4. In the Materials and Methods section, it is equal to 1, which makes it possible to summarize the occurrence of all pentapeptides that differ from the given pentapeptide by no more than one residue. This means that the pentapeptide is considered in a broader, evolutionary sense, with acceptable substitutions. At the large values of δ, for example, δ = 2 or δ = 3, the depth of the evolutionary perspective will increase, allowing for a greater number of substitutions in the pentapeptide. Having selected a separate branch of the hierarchical tree, it will be possible to consider the evolution of individual elements of the hierarchical structure by introducing for comparison a criterion similar to (1). This criterion will make it possible to compare protein sequences that are far apart in the process of evolution. 

The obtained results open up a possibility of creating a mathematically substantiated hierarchical classification of the structural elements of proteins. We plan to develop such a classification in the near future.

## Figures and Tables

**Figure 1 ijms-22-08339-f001:**
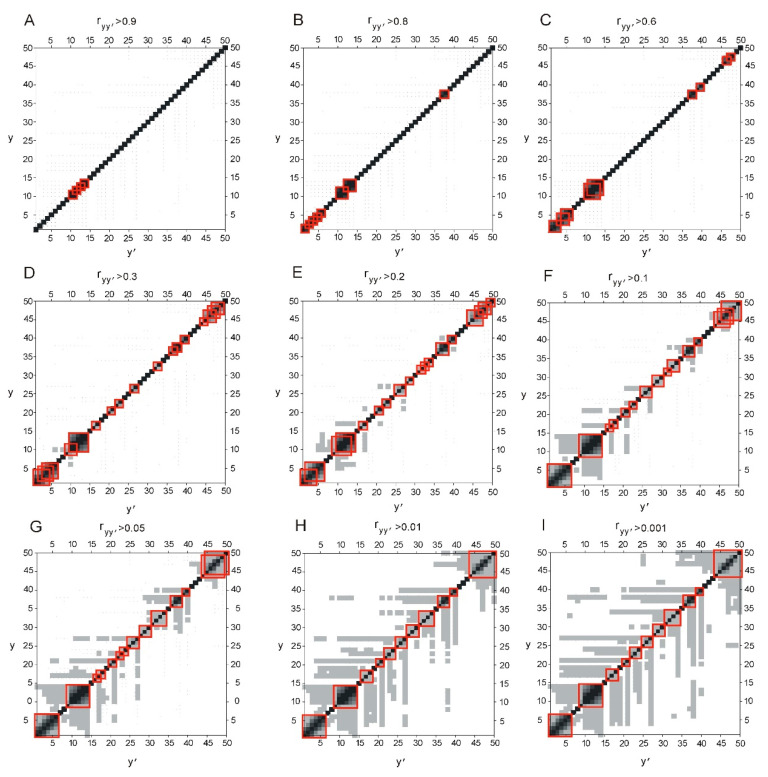
Graphical representation of the matrix $r_{yy{}^{\prime }}$. Matrix elements are displayed in the figure if they exceed the specified threshold value. Nine threshold values were used from 0.9 to 0.001 (**A**–**I**). All matrix elements which values $r_{yy{}^{\prime }}$ exceed threshold are displayed in gray. Matrix elements with values $r_{yy{}^{\prime }}$ equal to 1 are displayed in black. Matrix elements $r_{yy{}^{\prime }}$ (given by Equation (9)) reflect correlation of vectors $V_{y}$ of first derivatives $\overset{}{\tilde{S{{}^{\prime }}_{I}}(y)}$ for different thresholds.

**Figure 2 ijms-22-08339-f002:**
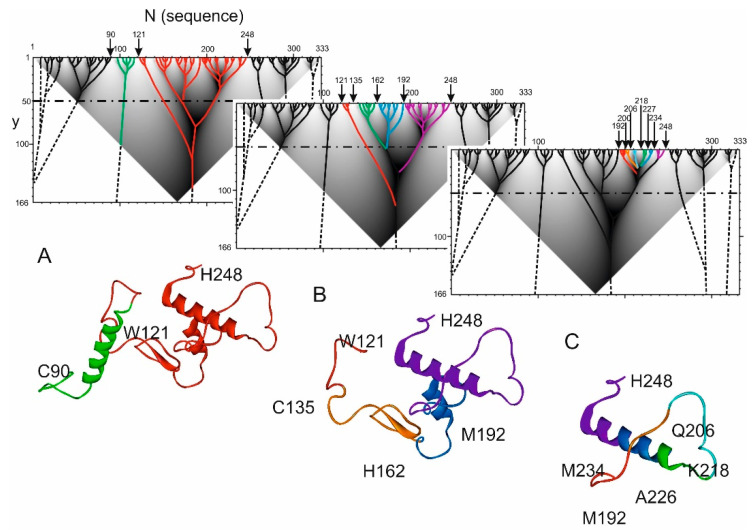
The correspondence between the hierarchical structure of the sequence and the spatial structure of photosynthetic reaction center protein from Rhodopseudomonas viridis fragments (PDB id 1PRC chain C). Dot-dash line marks y = 50. Dashed lines show the ELIS approximation beyond the boundary of the triangular definition area. (**A**) Hierarchical structure of the protein sequence with two colored high-rank ELISes: 90–121 (green) and 121–248 (red) with corresponding spatial structure. ELISes located at the N- and C-ends of the protein sequence are shown in black on the hierarchical structure. (**B**) Elements of middle-level hierarchical structure are shown in red (121–135), in green (135–162), in blue (162–192) and in violet (192–248) with corresponding spatial structure. (**C**) Elements of low-level hierarchical structure are shown in red (192–200), in orange (200–206), in cyan (206–218), in green (218–227), in blue (227–234) and in violet (234–248) with corresponding spatial structure.

**Figure 3 ijms-22-08339-f003:**
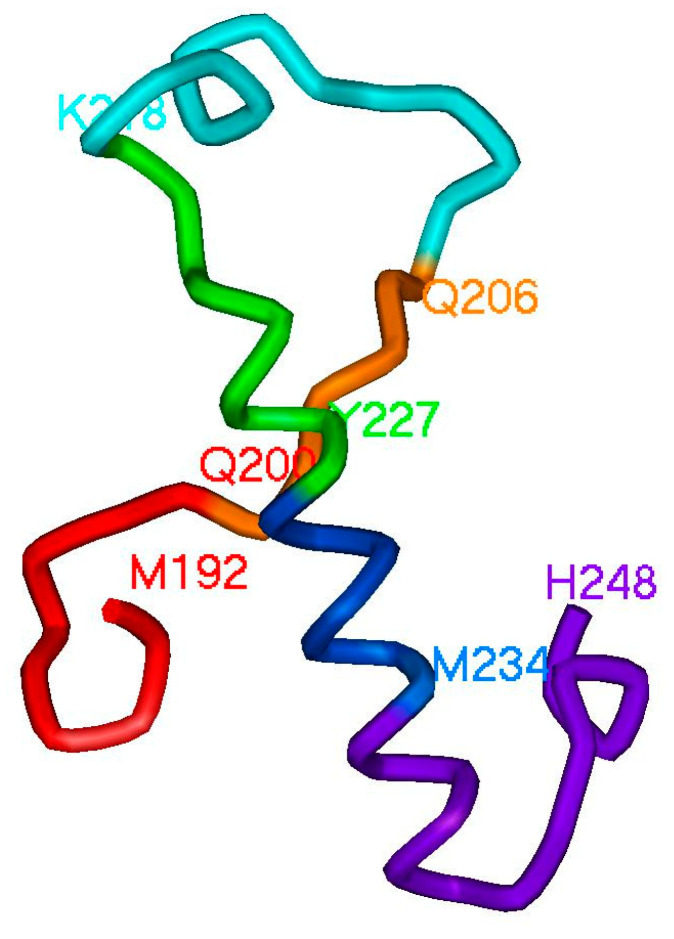
Division into elements at a lower level of the hierarchy the ELIS 192–248. Red (192–200), orange (200–206), blue (206–218), green (218–227), blue (227–234) and violet (234–248) elements of the hierarchical information structure are marked.

**Figure 4 ijms-22-08339-f004:**
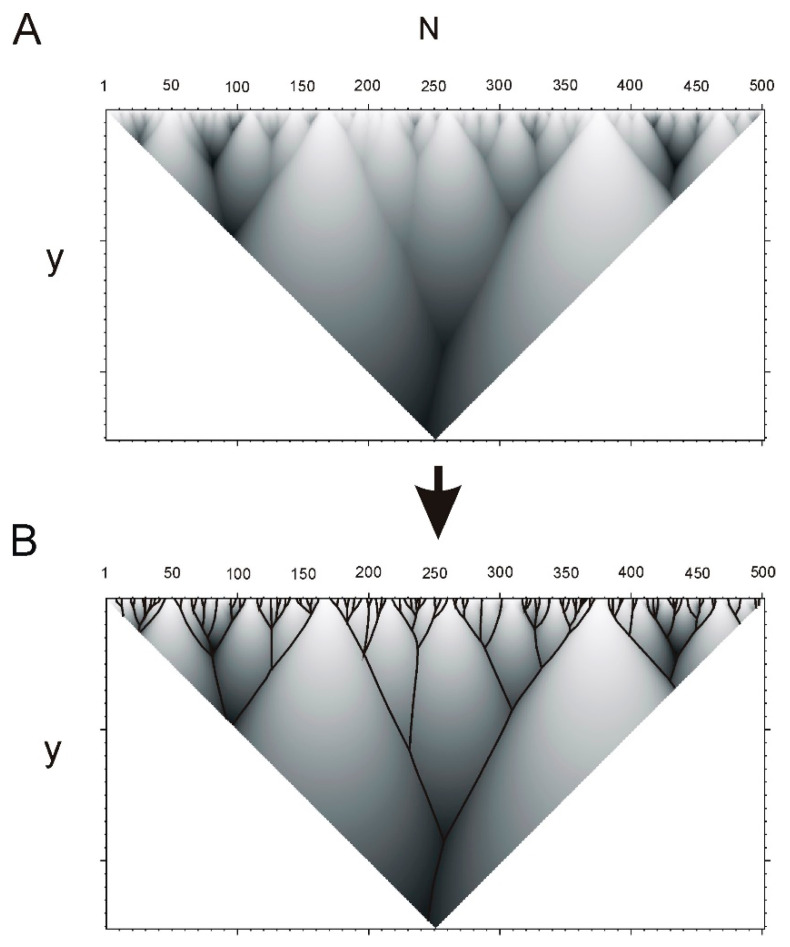
Example of hierarchical structure of protein sequence determined by function (4). (**A**). Hierarchical structure of the sequence of catalase protein (UniProt id P29422) obtained by the ANIS (analysis of hierarchical structure) method. (**B**). Tree-like graph is constructed using local maxima of the function $H_{I}(x,y)$ (4). ELIS are given by branches of the graph. Notations of axes: N—number of amino acid residue in the protein sequence, y—semi-width of the Gaussian function (2).

**Figure 5 ijms-22-08339-f005:**
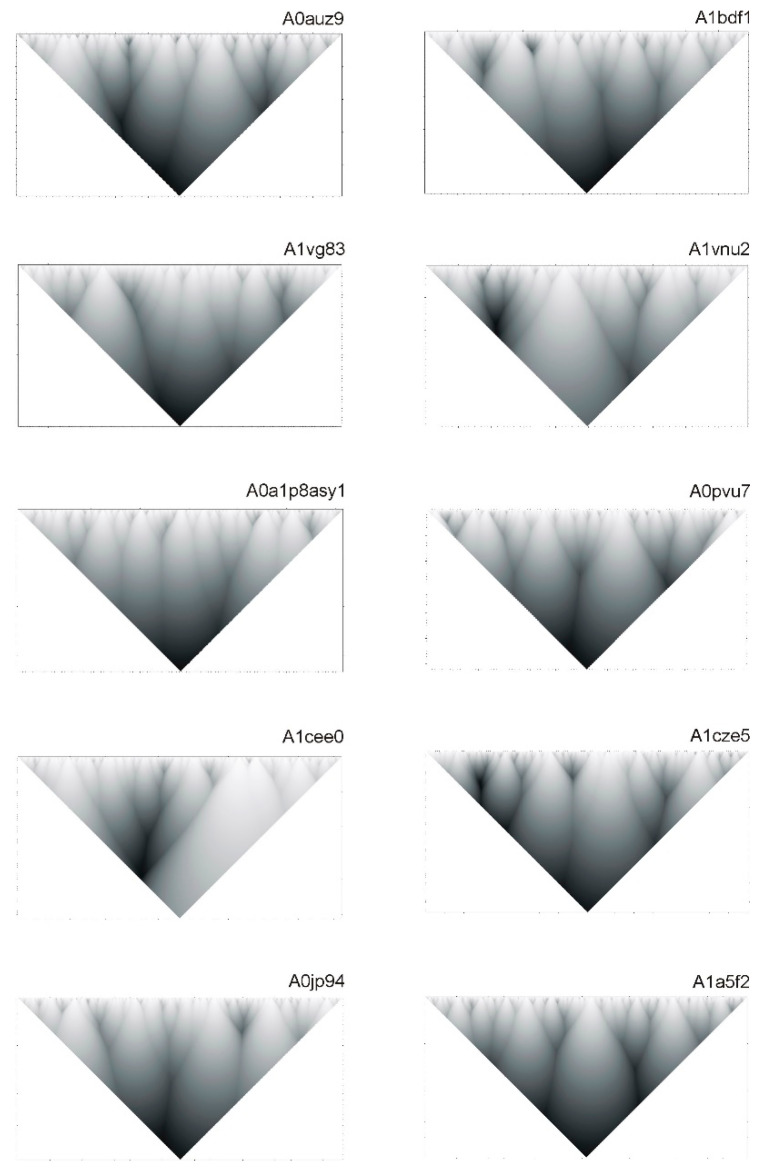
Hierarchical structures of some protein sequences. Codes of sequences from database UniProt are indicated above the pictures.

**Figure 6 ijms-22-08339-f006:**
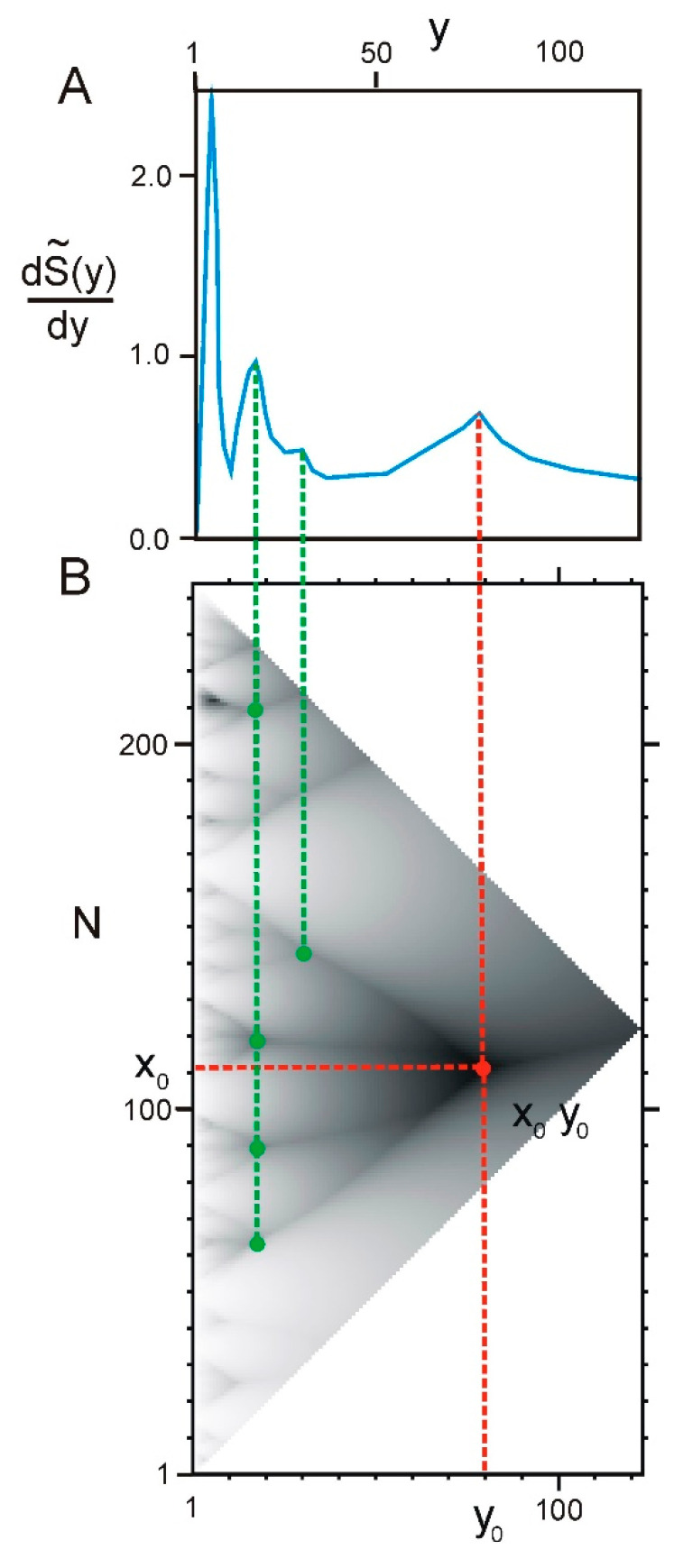
First derivative $\overset{}{\tilde{S{{}^{\prime }}_{I}}(y)}$ with respect to y. (**A**) Graph of the derivative of the difference between real and model entropy of partition (Equation (8)). Dashed line shows relation between points of branching in the hierarchical structure of protein sequence (**B**) and maxima at the graph of the first derivative (**A**). (**B**) Hierarchical structure of protein sequence of 26S proteasome regulatory subunit rpn10 (UniProt id O94444) where branching points are indicated. Branching point $x_{0}y_{0}$ is mentioned above and indicated by red. Other branching points are indicated by green.

**Table 1 ijms-22-08339-t001:** Characteristic values of smoothing function width value **y** for different CSA.

Continuous Square Areas, Index Number	L1	L2	L3	L4	L5	L6	L7	L8	L9	L10	L11
Range of smoothing function width value *y*	1–6	9–14	16–18	20–21	22–24	25–27	28–30	31–34	36–38	39–40	44–50

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
