# Peer review of "Hierarchical Structure of Protein Sequence"

_ijms, 2021, doi:10.3390/ijms22158339_

Round 1

Reviewer 1 Report

The authors have described a mathematical method for the identification of the hierarchical organization of protein sequences.  This method is based on pentapeptide as a unit of protein sequences. 

  • The authors should describe in more depth why this method is better than the method of Kaushik and Zhang (2020).
  • The discussion needs to be elaborated in the context of findings and previous literature.
  • How this method will be useful for the study of the molecular evolution of protein, and in the designing of recombinant proteins? 

Author Response

Review 1

The authors have described a mathematical method for the identification of the hierarchical organization of protein sequences.  This method is based on pentapeptide as a unit of protein sequences.

We thank the reviewer for the very positive grades and precise summary of our work.

The authors should describe in more depth why this method is better than the method of Kaushik and Zhang (2020).

In the work of Kaushik and Zhang (2020), the authors proposed a method for distinguishing between natural and random protein sequences, which takes into account the occurrence of amino acids and tripeptides. These authors did not offer methods for revealing hierarchy or isolating individual elements in a protein sequence. A more detailed explanation has been added to the Introduction section.

The discussion needs to be elaborated in the context of findings and previous literature.

The Discussion has been rewritten and expanded.

How this method will be useful for the study of the molecular evolution of protein, and in the designing of recombinant proteins?

Structural domains are stable structural elements and may independently form the 3D structure. Correspondence between the highest rank ELIS to structural domains may point to the important role of high rank ELIS in the formation of the spatial structure of proteins. We assume that the elements of 3D structure corresponding to the elements of information structure of other ranks are also stable elements of proteins 3D structure. This method is promising for use in the design of recombinant proteins, since the removal or addition of individual branches of the hierarchical tree should not affect the folding of the protein as a whole. Previously, this method was already used for the design of recombinant proteins [24,27] and solving biotechnological problems [22,23,25,26].

Our method also provides opportunities for studying the evolution of proteins. In this article, the value of δ in formula (1) in Section 4. Materials and Methods is equal to 1, which makes it possible to summarize the occurrence of all pentapeptides that differ from the given pentapeptide by no more than one residue. That is, the pentapeptide is considered in a broader, evolutionary sense, with acceptable substitutions. At the large values of δ, for example, δ = 2 or δ = 3, the depth of the evolutionary perspective will increase, allowing for a greater number of substitutions in the pentapeptide. Having selected a separate branch of the hierarchical tree, it will be possible to consider the evolution of individual elements of the hierarchical structure by introducing for comparison a criterion similar to (1). This criterion will make it possible to compare protein sequences that are far apart in the process of evolution.

Reviewer 2 Report

The submitted article on hierarchical structure analysis of proteins is interesting. However, I would like to make some comments on the text:
1) In the Introduction it is not clearly explained what the authors mean by levels of hierarchical structure. What are these levels? Appropriate drawing or scheme would be very helpful.
2) Lines 80-81: Was there any selection of selected proteins? What was the guiding principle in their selection? How diverse and representative was this group? Can we distinguish subgroups in this group and would similar analyses for them give similar results?
3) Fig. 1: Axis labels and titles are relatively small and hardly readable.
4) Abbreviations like "ELIS" appear in the text but are nowhere elaborated or explained. 
5) I would ask that the authors address the question of the practical significance of these analyses. In the Conclusions the perspectives of using the method are emphasized, but to what extent these levels of hierarchical structure distinguished by the Authors have a bearing on the structure of actual concrete molecules?

Author Response

Review 2

The submitted article on hierarchical structure analysis of proteins is interesting. However, I would like to make some comments on the text:

1) In the Introduction it is not clearly explained what the authors mean by levels of hierarchical structure. What are these levels? Appropriate drawing or scheme would be very helpful.

The hierarchy level in tree-like hierarchical graphs is determined by the maximum number of merge points when moving along the graph along the y-axis (see figure below). The Materials and Methods describe a method for constructing a hierarchical graph by protein sequence (see Figure 4B). Figure 2 shows the relationship between the levels of hierarchy in a graph with elements of spatial structure.

In the Introduction section and in the caption to Table 1, the corresponding descriptions were corrected. The concept of a hierarchical level is clarified in the text of the article.

2) Lines 80-81: Was there any selection of selected proteins? What was the guiding principle in their selection? How diverse and representative was this group? Can we distinguish subgroups in this group and would similar analyses for them give similar results?

Correlation matrix  was built for 24 647 protein sequences with sizes in between 50 to 400 of amino acid residues from the NRDB90 database [1].

  1. Holm L., Sander C. Removing near-neighbour redundancy from large protein sequence collections // Bioinformatics. 1998. Vol. 14, â„– 5. P. 423–429.

Diversity and representativeness of the NRDB dataset did not studied by us. This is a dataset of non-homologous proteins, and the selection of subgroups in it is a serious scientific task.

3) Fig. 1: Axis labels and titles are relatively small and hardly readable.

Figure 1 corrected.

4) Abbreviations like "ELIS" appear in the text but are nowhere elaborated or explained.

Corrected

5) I would ask that the authors address the question of the practical significance of these analyses. In the Conclusions the perspectives of using the method are emphasized, but to what extent these levels of hierarchical structure distinguished by the Authors have a bearing on the structure of actual concrete molecules?

In the Conclusion section, a description of the prospects for using the method for protein engineering and for studying protein evolution has been added.

The relationship between the elements of the hierarchical structure of the sequence and fragments of the protein structure is described in more detail in the Discussion.

Reviewer 3 Report

Dear authors,

I find it extremely intriguing to co-relate and come up with a generalized idea about sequence-structure co-relation between proteins. However, I do feel that authors have not shed light on why this is important and how these methods can be applied in various ways to tackle various biological problems.

Comments:

  • It is not clear to me the reason of choosing a pentapeptide as a unit of protein sequences.
  • Do authors have or plan to test these results by wet-lab experiments
  • I am also curious to know how the authors would approach this problem when dealing with membrane proteins.

Thank you!

Author Response

Review 3

Dear authors,

I find it extremely intriguing to co-relate and come up with a generalized idea about sequence-structure co-relation between proteins.

We thank the reviewer for the very positive grades and precise summary of our work.

However, I do feel that authors have not shed light on why this is important and how these methods can be applied in various ways to tackle various biological problems.

In order to emphasise importance and applicability of the method to various problems we expanded Introduction, Discussion and Conclusion sections.

Comments:

It is not clear to me the reason of choosing a pentapeptide as a unit of protein sequences.

The relationship between the amino acid sequence of a protein and its spatial structure seems intuitively quite obvious to us. We know that the amino acid sequences of proteins are encoded by the corresponding genes, and evolutionarily related organisms have similar sequences of genes and proteins that perform the same functions. That is, such protein sequences fold into spatially similar structures, especially in the region of active sites. Moreover, quite a large number of amino acid substitutions are often required to change the topology of the protein fold. There are examples where the topology of the fold is preserved when the sequence similarity is less than 30%. However, there are other examples when a single substitution leads to a change in protein folding and causes severe pathology.

Let's take a look at the process of protein synthesis on the ribosome. Protein folding essentially begins during synthesis, when the protein is only partially synthesized. Folding depends on local interactions of side radicals of amino acid residues of the protein chain coming together in space. Accordingly, for correct folding, local interactions between certain amino acid residues should occur at the right stage, which lead to the formation of the correct pre-folding conformation, i.e. some positions in the protein sequence must be correlated. We have previously shown in [1] that such a cross-correlation of positions in the amino acid sequence actually exists. We have shown that the greatest correlation is observed at small distances between amino acids in the sequence. The most optimal size was 5 amino acids [1,2].

  1. Nekrasov A.N. Entropy of Protein Sequences: An Integral Approach // Journal of Biomolecular Structure and Dynamics. 2002. Vol. 20, â„– 1. P. 87–92.
  2. Nekrasov A.N., Anashkina A.A., Zinchenko A.A. A new paradigm of protein structural organization // Proc. 2nd Int. Conf.“Theoretical Approaches to Bioinformation Systems”(TABIS 2013). Institute of Physics, Belgrade, 2014. P. 1–22. https://www.researchgate.net/profile/Alexei-Nekrsov/publication/270684947_A_New_Paradigm_of_Protein_Structural_Organization/links/54b285da0cf220c63cd25c21/A-New-Paradigm-of-Protein-Structural-Organization.pdf#page=15

Do authors have or plan to test these results by wet-lab experiments

This method has already been successfully applied in wet-lab experiments on bioengineering and design of recombinant proteins, for example

  1. A.N. Nekrasov, V.V. Radchenko, T.M. Shuvaeva, V.I. Novoselov, E.E. Fesenko, V.M. Lipkin ”The Novel Approach to the Protein Design: Active Truncated Forms of Human 1-CYS Peroxiredoxin” J. Biomol. Struct. Dyn. (2007) vol. 24(5), pp.455-462.
  2. Nekrasov A.N., Petrovskaya L.E., Toporova V.A., Kryukova E.A., Rodina A.V., Moskaleva E.Y., Kirpichnikov M.P. ”Design of a novel interleukin-13 antagonist from analysis of informational structure” Biochemistry (Mosc). 2009 vol. 74(4), pp. 399-405.
  3. Yves Briers, Konstantin Miroshnikov, Oleg Chertkov, Alexei Nekrasov, Vadim Mesyanzhinov, Guido Volckaert, Rob Lavigne ”The structural peptidoglycan hydrolase gp181 of bacteriophage KZ” Biochem. Biophys. Res. Commun. (2008) vol. 374(4), pp. 747-751.
  4. Shingarova L.N., Petrovskaia L.E., Nekrasov A.N., Kriukova E.A., Boldyreva E.F., Iakimov S.A., Gur’ianova S.V., Dolgikh D.A., Kirpichnikov M.P. ”Production and properties of human tumor necrosis factor peptide fragments” Bioorg Khim. 2010 vol. 36(3), pp. 327-336 Russian.

[15] [A.N. Nekrasov, A.. Zinchenko ”Structural Features of the Interfaces in EnzymeInhibitor Complexes” J. Biomol. Struct. Dyn. (2010) vol. 28(1), pp. 85-96

I am also curious to know how the authors would approach this problem when dealing with membrane proteins.

Hierarchical structure for membrane proteins can be calculated in the same way. Below is a hierarchical structure of the human Na,K-ATPase alpha-1 subunit. Description of the sequence and structure you can find at https://www.uniprot.org/uniprot/P05023

Round 2

Reviewer 2 Report

Thank you for the responses to my comments.